# From Continual Learning to Causal Discovery in Robotics [*]

**Luca Castri, [1] Sariah Mghames, [1] Nicola Bellotto [1,2]**

[1]University of Lincoln, UK
[2]University of Padua, Italy
{lcastri, smghames}@lincoln.ac.uk, nbellotto@dei.unipd.it

## Abstract

Reconstructing accurate causal models of dynamic systems from time-series of sensor data is a key problem in many real-world scenarios. In this paper, we present an overview based on our experience about practical challenges that the causal analysis encounters when applied to autonomous robots and how Continual Learning (CL) could help to overcome them. We propose a possible way to leverage the CL paradigm to make causal discovery feasible for robotics applications where the computational resources are limited, while at the same time exploiting the robot as an active agent that helps to increase the quality of the reconstructed causal models.

## 1    Introduction

Causal discovery approaches generally build the causal model of the observed scenario from static or time-series data collected and processed in advance. However, in many real-world robotics applications, this approach could result inefficient or even unfeasible. The link between *Continual Learning* (CL) (Lesort et al. 2020) and *Causality* might represent a stepping stone towards the exploitation of causal discovery algorithms (Glymour, Zhang, and Spirtes 2019) that currently suffer many limitations in autonomous robots.

Causal inference is an active research area in different fields, including robotics and autonomous systems (Hellström 2021; Brawer, Qin, and Scassellati 2020; Cao et al. 2021; Katz et al. 2018; Angelov, Hristov, and Ramamoorthy 2019). However, most of these works overlooked some key features that are important for real-world application, i.e. the computational cost and the memory needed by causal analysis when long time-series are processed to reconstruct a causal model of the observed scenario. To this end, the CL's ability to enable the acquisition of more knowledge by trained models without forgetting previous information, and without using previous data recordings, might help to address these problems and to achieve better result in terms of quality of the causal analysis. For instance, a robot in an automated warehouse with humans and various objects (e.g. see Fig. 1) could observe and intervene in the interactions among them (e.g. worker and shelf) in order to build

a causal model and therefore a deep understanding of the situation. Since the limited hardware resources though, the robot's causal analysis might be slow and based on a limited amount of data, leading to a low quality causal model. The solutions suggested in this paper would allow the robot to overcome its hardware limitations and, moreover, to improve the quality of the causal models by continually feeding new data for causal analysis, discarding the old collected one. This would enable a more efficient use of the robot's memory and computing's resources compared to existing causal discovery's approaches. To summarise, this paper proposes a *Causal Robot Discovery* (CRD) approach to overcome current limitations in causal analysis for real-world robotics applications, addressing in particular:

- the computing and memory hardware resources of the robot, which may hinder its capability to perform meaningful causal analysis;
- the update of previous causal models with new observational and interventional data from the robot to generate more accurate ones.

The paper is structured as follows: related work about continual learning and causal discovery are presented in Section 2; Section 3 introduces our CRD approach and explains how the integration of continual learning could help to overcome the challenges of causal discovery in robotics; finally, we conclude the paper in Section 4 discussing our current and future work in this area.

## 2    Related Work

**Causal discovery:**    Several methods have been developed over the last few decades to derive causal relationships from observational data, which can be categorized into two main classes (Glymour, Zhang, and Spirtes 2019). The first one includes *constraint-based methods*, such as Peter and Clark (PC) and Fast Causal Inference (FCI), which rely on conditional independence tests as constraint-satisfaction to recover the causal graph. The second one includes *score-based methods*, such as Greedy Equivalence Search (GES), which assign a score to each Directed Acyclic Graph (DAG) and perform a search in this score space. More recently, reinforcement learning-based methods have also been used to discover causal structure (Zhu, Ng, and Chen 2020). However, many of these algorithms work only with static data

---

[*]This work has received funding from the EU H2020 research & innovation programme – grant agreement 101017274 (DARKO).

(i.e. no temporal information) and are not applicable to time-series of sensor data in many robotics applications, for which time-dependent causal discovery methods are instead necessary. To this end, a variation of the PC algorithm, called PCMCI, was adapted and applied to time-series data (Runge 2018; Runge et al. 2019; Saetia, Yoshimura, and Koike 2021).

**Causal robotics:** Causal inference has been recently considered in robotics, for example to build and learn a Structural Causal Model (SCM) from a mix of observation and self-supervised trials for tool affordance with a humanoid robot (Brawer, Qin, and Scassellati 2020). Other applications include the use of PCMCI to derive the causal model of an underwater robot trying to reach a target position (Cao et al. 2021) or to predict human spatial interactions in a social robotics context (Castri et al. 2022). Further causality-based approaches can be found in the robot imitation learning and manipulation area (Katz et al. 2018; Angelov, Hristov, and Ramamoorthy 2019; Lee et al. 2021). However, all these solutions rely on a fixed set of time-series for causal analysis and do not consider the computational cost and complexity for online update of the robot's causal models.

**Continual learning:** The concept of learning continually from experience has been present in artificial intelligence since early days (Weng et al. 2001). Recently this has been explored more systematically in machine learning (Hadsell et al. 2020; Parisi et al. 2019) and robotics (Lungarella et al. 2003; Lesort et al. 2020; Churamani, Kalkan, and Gunes 2020). To our knowledge though, few applications of the continual learning paradigm can be found in the causality field. Javed, White, and Bengio (2020) incorporate causality and continual learning with an online algorithm that continually detects and removes spurious features from a causal model. In (Kummerfeld and Danks 2012, 2013; Kocacoban and Cussens 2019, 2020), instead, algorithms for online causal structure learning are presented to deal with non-stationary data. This is a key feature of data from real-world environments, which is still under-investigated in robotics and therefore motivates our approach proposed next.

## 3 Causal Robot Discovery

A review of the literature revealed that the possible limitations of autonomous robots doing causal discovery with their own on-board sensors have not been taken into account. Indeed, the computational and memory requirements for long time-series of sensor data are often very demanding, making the use of previous algorithms for causal inference unfeasible on such platforms.

Our approach is partially inspired by the works of Kocacoban and Cussens (2019) for handling non-stationary data, but differs from it in two ways. First of all, we adopt the current state-of-the-art PCMCI method for causal discovery from time-series data; second, we propose to re-learn the causal model not only when the observed scenario changes, but also at each new robot's set of observations/interventions (periodically, e.g. every few minutes). In particular, the introduction of the CL paradigm could help the robot to overcome the challenge of limited hardware resources and to

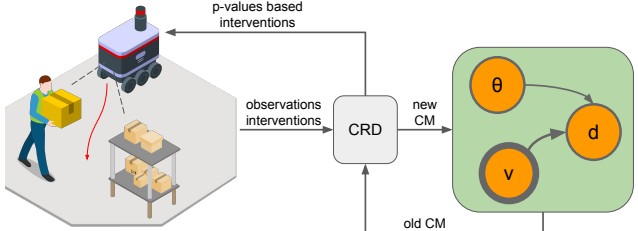

Figure 1: CRD approach: the robot provides observational and interventional data about human-object interactions to the CRD block. The latter generates a causal model, which is stored and used to compare the next one built on subsequent robot's observations and interventions. Based on the p-values of the previous causal graph's links, the CRD could suggest the robot which links need to be better tested by future interventions.

improve the quality of the causal analysis even with non-stationary data. In addition, a CRD approach could benefit from the fact that robots are physically embodied in the environment and can actively influence its dynamic processes (i.e. by performing interventions). That is, CRD could improve the accuracy of the causal model by enriching "passive" observational data from the sensors with "active" interventional data from robot's actions aimed at collecting specific time-series for causal discovery.

Therefore, our goal is to decrease the need of hardware resources – often scarce in autonomous systems – and to increase the quality of the causal analysis by using the robot as an active agent in the learning process. The use of CRD would allow the creation of high quality causal models by continually updating them with new sensor data from robot's observations and interventions, without the need to re-process the whole time-series but only new information in an incremental fashion. The CRD system envisaged in this paper is thought to limit the demand for hardware resources and allow the robot to perform high quality causal discovery in a reasonable time by using its own on-board sensor data.

The proposed approach is depicted in Fig. 1: (*i*) starting from a prefixed set of variables, the robot collects meaningful data by observing and intervening in the target scenario; (*ii*) based on this data, a causal model is estimated using PCMCI (Runge 2018), which computes test statistics and p-values as causal strengths of the DAG's links. At this stage, differently from (Kocacoban and Cussens 2019), to increase the accuracy of the causal discovery, the robot keeps on collecting data by observing and intervening in the scenario to create new causal models. Periodically then, the robot compares the new causal models with the old ones, inheriting from the latter only the links that minimise the p-values of the DAG's causal relations. By repeating this process until the observed scenario changes, a stable version of the causal model with minimum uncertainty levels would be reached. This is useful not only for modeling the current scenario, but also when it changes. Indeed, by re-using part of the stored causal model for an initial scenario, the new causal

discovery when the scenario changes can be significantly sped up (Kocacoban and Cussens 2019).

Note that by iteratively discarding the old time-series and storing only the built causal model helps to avoid the combinatorial explosion otherwise affecting PCMCI, therefore allowing the robot to operate and compute new models within reasonable time. Furthermore, the catastrophic-forgetting problem is mitigated by the fact that, during continuous operations, the robot observes similar processes with only small incremental changes, which leads to sequences of similar causal models reconstructed from relatively small variations of previous ones.

The operations performed by our CRD approach are represented by the flowchart in Fig. 2 and described next.

1. The process starts with a first set of observational data (i.e. sensors' time-series) collected by the robot. An *inference matrix* is estimated by performing conditional independence tests (e.g. correlation, transfer entropy) on the time-series, producing an initial causal model.

2. Afterwards, since at the first attempt there are no stored causal models, the "Interventions Strategy" (red block in Fig. 2) is initially neglected and the "Causal Model Optimisation" (blue block) is used only to estimate and save the causal model (CM), together with its test statistics and p-values matrices.

3. Once the causal model of the observed scenario is obtained, the robot can improve its quality by providing new data to the CRD. In practice, the robot collects new time-series of sensor data from the scenario so that a new inference matrix can be estimated. At this stage, two parallel processes are executed.

   - The first one, "Interventions Strategy", compares the CM obtained at the previous iteration with the inference matrix just estimated. If the stored CM still fits the estimated inference matrix, it means we are in a stationary-data case and the robot is observing the same scenario of the previous iteration. Therefore, based on the p-values matrix of the stored CM, the CRD might suggest the most "unrelaible" links that need to be re-checked. These are used by the robot to plan the next interventions.

   - The second process, "Causal Model Optimisation", performs a fresh causal discovery on the new data and compare the obtained CM with the one previously stored. From this comparison, a new CM is derived that inherits only the links minimising the p-values of the causal graph. The result is then stored to be used at the next iteration.

In case of stationary data, by repeating this procedure, a stable version of the causal model with minimum uncertainty can be reached. In case of non-stationary data, new time-series will be provided to the CRD, which will detect any significant variations by comparing the newly estimated inference matrix with the one of the stored CM. In this case there is no Interventions Strategy; instead, the Causal Model Optimisation reconstructs the new model exploiting the similarities with the stored CM to speed up the analysis. In par-

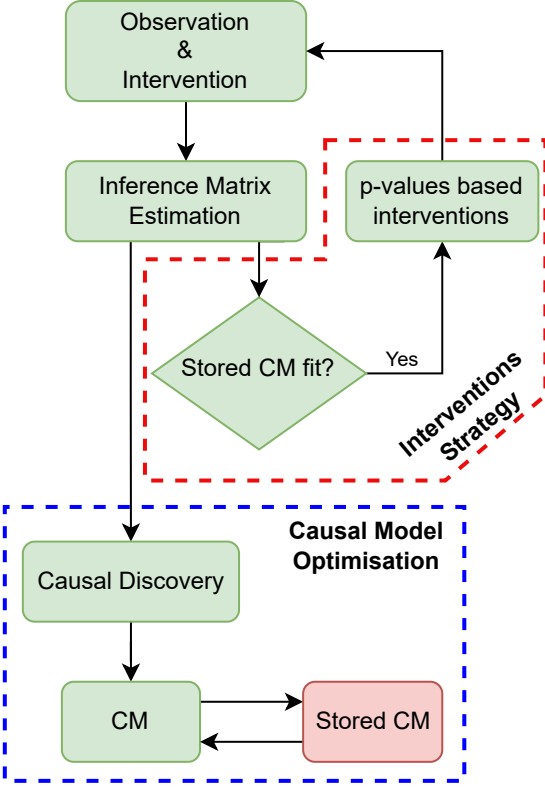

Figure 2: CRD flowchart.

ticular, the last step is performed by comparing the new inference matrix with the old CM's one in order to identify previous causal links that are still valid for the new model.

## 4 Conclusion

In this paper we considered the hardware resource limitations of autonomous robots, which are crucial to perform causal inference, and proposed a new approach for causal robot discovery to overcome some of the main challenges. This includes improving the quality of the causal models by using the robot as an active agent in the learning process.

To summarise, in both stationary and non-stationary data cases, the CRD discards the time-series data after each new CM reconstruction, allowing the robot to perform causal discovery within reasonable time. Moreover, as already explained, in case of non-stationary data the old CM can be partially exploited to speed up the reconstruction of the new one. This favors not only the execution time of the causal analysis but also the handling of catastrophic-forgetting phenomena. Indeed, in case of non-stationary data, assuming small and incremental variations of the observed scenario, the new causal model is reconstructed by partially exploiting the old one, thus reducing the possibility of completely forgetting what was previously learnt.

Future work will be devoted to the implementation and application of this approach to real-wold robotics problems, with a special interest in industrial scenarios involving human-robot interaction and collaboration.

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
