# OpenReview forum: "From Continual Learning to Causal Discovery in Robotics"
_AAAI.org/2023/Bridge/CCBridge — AAAI23 Bridge Continual Causality_

### Official Review · Reviewer_uByV · 2022-12-01

**Rating:** 7
**Confidence:** 4

**Review:**

This paper describes Causal Robot Discovery (CRD), a continual learning approach for causal discovery. The approach can address current challenges in limited compute resources as well as efficient update of causal models continuously.

The paper gave a clear overview of their proposed method, which is good for applying continual learning to causal inference. The method also has potential applications in important areas such as warehouse robots.

---

### Official Review · Reviewer_fKJn · 2022-12-02
**Proposes to link continual learning and causal discovery in robotic settings with limited resources**

**Rating:** 5
**Confidence:** 3

**Review:**

The short paper proposes that causal robot discovery, with stringent hardware resource limitations, will require continual learning. A robot provides continual observational and interventional data about human-object interactions to a causal robotic discovery module whose output is monitored and adapted over time.  The PCMCI method for causal discovery
from time-series data is used and it is proposed to re-learn the causal model under observed scenario changes as well as at each new robot’s set of observations/interventions (periodically, e.g. every few minutes).

Pros: Clarity of paper:  ok, Background Literature: ok
Cons:  Novelty is minimal, as it is an Incremental extension

---

### Official Review · Reviewer_Lnxf · 2022-12-02
**Brideing the gap between CL to Casual Discrovery in Robotics**

**Rating:** 5
**Confidence:** 2

**Review:**

This paper presents the casual discovery that integrates the continual learning paradigm in robotics.
The paper clearly states why the notion of continual learning in casual discovery could be practically helpful in robotics.

The research background and paradigm are interesting. However, it is hard to capture how the continual learning method contributes to casual discovery. The paper states the general flow of the casual discovery method with a brief mention that continual learning can overcome the streaming situation and hardware limits. However, this needs to clearly state how the continual learning methods overcome the issue of the aforementioned problem, and how catastrophic forgetting might impact casual discovery.

Overall, the research question and practical settings are adequate and exciting. However, with the current form, it's hard to see the clear impact.

---

### Decision · Program_Chairs · 2022-12-05

**Decision:**

Accept

**Comment:**

Accept - Poster

This paper discusses the hardware limitations of autonomous robots and proposes a novel approach for causal discovery by leveraging continual learning. The paper meets the overall themes of the bridge program well and would seemingly catalyze a lot of interesting discussion surrounding practical applications of bridging both fields. We strongly suggest that the authors integrate the reviewers' comments for the camera-ready version of the paper. For example, the authors should substantiate further upon how continual learning will specifically improve causal discovery.